# Specificity in endoplasmic reticulum-stress signaling in yeast entails a step-wise engagement of *HAC1* mRNA to clusters of the stress sensor Ire1

Eelco van Anken[1,2]*, David Pincus[2†], Scott Coyle[3], Tomás Aragón[2,4], Christof Osman[2], Federica Lari[1], Silvia Gómez Puerta[4], Alexei V Korennykh[2‡], Peter Walter[2]

[1]Division of Genetics and Cell Biology, San Raffaele Scientific Institute, Milan, Italy; [2]Department of Biochemistry and Biophysics, Howard Hughes Medical Institute, University of California, San Francisco, San Francisco, United States; [3]Department of Biochemistry and Biophysics, University of California, San Francisco, San Francisco, United States; [4]Department of Gene Therapy and Gene Regulation, Center for Applied Medical Research, Pamplona, Spain

**\*For correspondence:**
evananken@mac.com

**Present address:** [†]Whitehead Institute for Biomedical Research, Cambridge, United States; [‡]Department of Molecular Biology, Princeton University, Princeton, United States

**Competing interests:** The authors declare that no competing interests exist.

**Abstract** Insufficient protein-folding capacity in the endoplasmic reticulum (ER) induces the unfolded protein response (UPR). In the ER lumen, accumulation of unfolded proteins activates the transmembrane ER-stress sensor Ire1 and drives its oligomerization. In the cytosol, Ire1 recruits *HAC1* mRNA, mediating its non-conventional splicing. The spliced mRNA is translated into Hac1, the key transcription activator of UPR target genes that mitigate ER-stress. In this study, we report that oligomeric assembly of the ER-lumenal domain is sufficient to drive Ire1 clustering. Clustering facilitates Ire1's cytosolic oligomeric assembly and *HAC1* mRNA docking onto a positively charged motif in Ire1's cytosolic linker domain that tethers the kinase/RNase to the transmembrane domain. By the use of a synthetic bypass, we demonstrate that mRNA docking per se is a pre-requisite for initiating Ire1's RNase activity and, hence, splicing. We posit that such step-wise engagement between Ire1 and its mRNA substrate contributes to selectivity and efficiency in UPR signaling.

## Introduction

Proteins that travel along the secretory pathway first enter the lumen of the endoplasmic reticulum (ER) as unfolded polypeptides. Assisted by ER-resident enzymes, they undergo oxidative folding, modification, and assembly reactions. When properly folded, they are packaged into ER exit vesicles and travel to their final destination in the endomembrane system, on the cell surface or, after secretion, outside of the cell. Proteins that do not reach maturity are degraded by the proteasome after retrotranslocation into the cytosol (via ER-associated degradation) or by autophagy (*Ellgaard and Helenius, 2003*; *van Anken and Braakman, 2005*; *Bernales et al., 2006*). Homeostasis in ER protein folding is achieved by fine-tuning the balance between the protein folding load and the protein folding capacity in the ER lumen (*Mori, 2009*; *Kimata and Kohno, 2011*; *Walter and Ron, 2011*).

In yeast, Ire1 is the only known ER-stress sensor that responds to an accumulation of misfolded proteins in the ER lumen and transduces this information across the ER membrane. On the cytosolic side, Ire1 activation results in the non-conventional splicing of *HAC1* mRNA, which is cleaved by Ire1's RNase domain at two splice sites, releasing a single intron (*Sidrauski and Walter, 1997*). Upon ligation of the severed exons, the spliced mRNA is translated to produce the Hac1 transcription activator that drives expression of UPR target genes to mitigate ER-stress (*Travers et al., 2000*; *Walter and Ron, 2011*).

**eLife digest** Proteins are built based on instructions in template molecules called messenger RNAs (or mRNAs), which are copied from the DNA of genes. As they are made, proteins must fold into a specific three-dimensional shape and some proteins pass into a compartment in the cell, called the endoplasmic reticulum, in which they fold. So-called molecular chaperone proteins assist this folding process. From the endoplasmic reticulum, most proteins travel to other destinations within or outside of the cell.

If the molecular chaperones in the endoplasmic reticulum are overwhelmed by their protein folding task, unfolded proteins accumulate; a situation that can be harmful to the cell. In eukaryotic cells including yeast, a sensor protein called Ire1 detects when unfolded proteins build up in the endoplasmic reticulum. As a result, the Ire1 sensor proteins join together to form clusters and an mRNA molecule called *HAC1* is specifically recruited to the Ire1 clusters. The portions of the Ire1 protein that extend out from the endoplasmic reticulum into the cell proper then bind to *HAC1* mRNA and cut a piece out of it. This edited mRNA encodes the instructions to build a protein that in turn boosts the expression of various components—including the appropriate molecular chaperones—that are needed to alleviate the stress caused by an excess of unfolded proteins.

Within clusters, individual Ire1 proteins interact through the portions of the protein found on the inside of the endoplasmic reticulum. Now, van Anken et al. show that these interactions are sufficient for forming and maintaining clusters. The interactions between the portions of the Ire1 proteins outside of the endoplasmic reticulum are needed for editing the *HAC1* mRNA but not for forming and maintaining the clusters or for recruiting the *HAC1* mRNA molecule to bind to Ire1. Instead, van Anken et al. discovered an mRNA binding site on the Ire1 clusters, which is separate from the part of the Ire1 protein that cuts the mRNA molecules. The Ire1 protein needs to first bind the *HAC1* mRNA molecule at this binding site before it can cut it; van Anken et al. suggest that this two-step process helps ensure accurate and efficient editing of the *HAC1* mRNA by Ire1. This process could also help to minimize the chance of other mRNA molecules being edited by mistake.

It will be of interest to investigate if similar safety measures are key for endoplasmic reticulum stress signaling mechanisms in humans, and whether these newly discovered steps can be targeted by drugs to treat disease.

Ire1 is activated through higher-order oligomerization (*Kimata et al., 2007*; *Aragón et al., 2009*; *Korennykh et al., 2009*). Two ER-lumenal domain (LD) interfaces, IF1$^L$ and IF2$^L$ ('L' for lumenal), which were identified in the crystal structure of yeast Ire1 LD and validated by mutagenesis, mediate oligomeric assembly of the LD (*Credle et al., 2005*; *Kimata et al., 2007*; *Aragón et al., 2009*; *Gardner and Walter, 2011*). Dimerization via IF1$^L$ yields a composite groove extending across the LDs of two Ire1 molecules (*Credle et al., 2005*). Unfolded stretches of polypeptides bind within this groove of Ire1 LD, causing its oligomerization in vitro (*Gardner and Walter, 2011*). Proximal activation of the UPR in vivo coincides with the dissociation of Kar2 (the yeast homolog of the ER-lumenal hsp70 chaperone BiP) from Ire1 (*Kimata et al., 2004*; *Pincus et al., 2010*; *Walter and Ron, 2011*). Yet, Ire1 mutants with impaired Kar2 binding still respond to ER-stress, although the threshold for activation is lowered (*Kimata et al., 2004*; *Pincus et al., 2010*; *Walter and Ron, 2011*). Thus, misfolded proteins likely are direct ligands that activate Ire1, while Kar2 fine-tunes the signaling (*Pincus et al., 2010*; *Gardner and Walter, 2011*).

On the cytosolic side of the ER membrane, Ire1 contains both a kinase and an RNase domain, which are tethered to the transmembrane domain via a linker (*Mori, 2009*; *Kimata and Kohno, 2011*; *Walter and Ron, 2011*). Three cytosolic assembly interfaces, IF1$^C$, IF2$^C$, and IF3$^C$ ('C' for cytosolic), were identified from the crystal structures of Ire1 kinase/RNase oligomers. IF1$^C$ creates back-to-back dimers of the kinase/RNase domains (*Lee et al., 2008*; *Korennykh et al., 2009*) that stack onto each other with an axial rotation via IF2$^C$ and IF3$^C$ to form filaments with a helical arrangement of seven Ire1 dimers per turn (*Korennykh et al., 2009*; *Walter and Ron, 2011*). The lumenal and cytosolic domain filaments predicted by the crystal structures have a different pitch and thus for steric reasons cannot be collinear. Instead, a two-dimensional arrangement of the two filaments, featuring ~20–30 Ire1 molecules, provides a model for the higher-order assembly in vivo (*Korennykh et al., 2009*; *Figure 1B*). This model

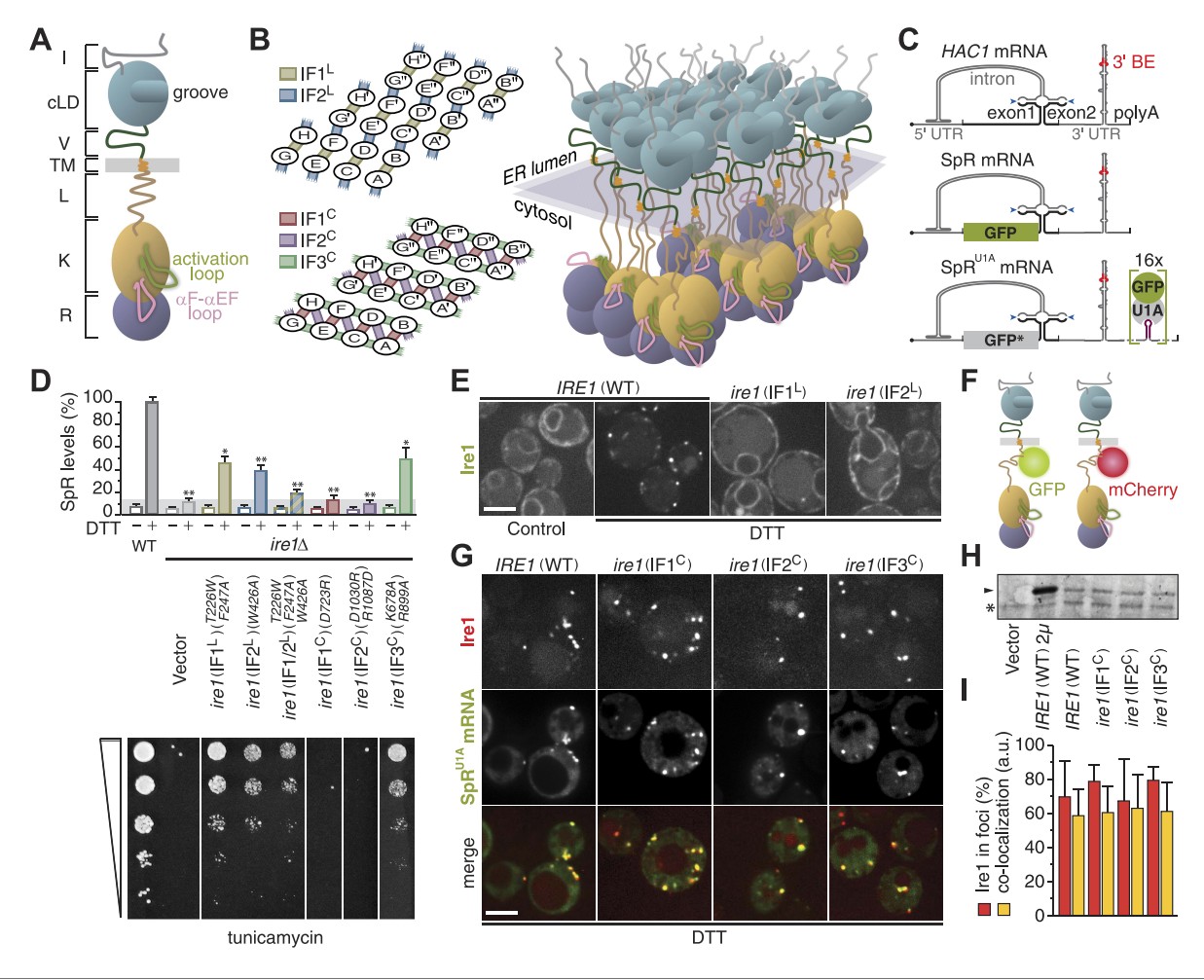

**Figure 1**. Oligomerization of Ire1's cytosolic domain is required for UPR signaling but not for Ire1 cluster formation or *HAC1* mRNA recruitment. (**A**) Schematic of *S. cerevisiae* Ire1. The ER-lumenal portion of Ire1 is divided in an N-terminal domain (I, gray), a core lumenal—ER-stress-sensing—domain (cLD, light blue), and BiP binding domain (V, dark green), which is tethered via a transmembrane (TM, orange) stretch to Ire1's cytosolic portion that is composed of a linker (L, brown), a kinase (K, ochre), and an RNase (R, purple) domain (***Walter and Ron, 2011***). The activation loop (light green) and the αF–αEF (pink) loop protrude from the kinase domain (***Lee et al., 2008***; ***Korennykh et al., 2009***). (**B**) A model architecture of a 24mer Ire1 cluster after oligomerization on either side of the ER membrane. Left: oligomerization via ER-lumenal interfaces IF1[L] (tan) and IF2[L] (steel blue) (top) and via cytosolic interfaces IF1[C] (indian red), IF2[C] (sea green), and IF3[C] (plum) (bottom). The 24 Ire1 molecules are labeled (A–H) A'–H', and A''–H''. IF1[C]-mediated back-to-back dimers are between A & B and C & D, etc. IF2[C], is formed between Ire1 molecules A and D, C and F, and so on. The third interface, IF3[C], is stabilized by a phosphate in the activation loop resulting from Ire1 trans-autophosphorylation. Dimerization via IF3[C] is therefore directional from B → D → F and from E → C → A, etc. (***Korennykh et al., 2009***). Right: three-dimensional rendering of the same 24 Ire1 molecules colored as in (**A**). (**C**) Top: schematic of *HAC1* mRNA. The *HAC1* open reading frame (ORF) is divided into two exons (black). The intron (gray) base pairs with the 5' UTR (gray), causing stalling of ribosomes. Ire1 cleaves the intron at the splice sites indicated by blue arrowheads. The 3' UTR (gray) harbors a stem-loop structure with the 3' BE (red) that facilitates recruitment of the *HAC1* mRNA to Ire1 foci (***Aragón et al., 2009***). The 5' m7G cap (•) and polyadenylation (polyA) signal are indicated. Middle: the green bar depicts the GFP ORF (green) that replaces part of the *HAC1* sequence in the splicing reporter (SpR). Translation of GFP only occurs when the intron is spliced from the mRNA, because removal of the intron by Ire1's endonuclease activity lifts a translational block caused by base pairing between the intron and the 5' UTR (***Pincus et al., 2010***). Bottom: 16 U1A binding sites (violet) were inserted into the 3' UTR of the SpR mRNA, bearing the non-fluorescent GFP-R96A mutant (GFP*, gray), downstream of the 3' BE containing stem-loop. Binding of GFP-tagged U1A protein allows visualization of the mRNA (***Aragón et al., 2009***). (**D**) Wild-type (WT) or *ire1Δ* cells, having a genomic copy of the SpR, were complemented with centromeric empty vector or bearing ire1 IF mutant alleles (***Aragón et al., 2009***; ***Korennykh et al., 2009***) as indicated. Top: SpR assay of cells. GFP fluorescence for 10,000 cells was measured by FACS analysis before or after ER-stress induction with 2 mM DTT for 2 hr, as described (***Pincus et al., 2010***); mean and s.d. are shown (n = 2). Bar diagrams for IF mutants are color-coded as in (**B**) left. The signal of DTT treated WT was set at 100%, while the signal reached in DTT treated *ire1Δ* cells due to auto-fluorescence of 14% was set as background (light gray bar). Statistical significance in a *Student's t-test* of differences in splicing levels as compared with wild-type is indicated (*p ≤ 0.05; **p ≤ 0.01). Bottom: viability

*Figure 1. Continued on next page*

*Figure 1. Continued*

assay by 1:5 serial dilutions spotted onto solid media with 0.2 μg ml$^{-1}$ of the ER-stress-inducer tunicamycin. Plates were photographed after 2–3 days at 30°C. (**E**) Localization of Ire1–GFP WT or IF$^L$ mutants before (left panel, control) and after (right panels, DTT) induction of ER-stress. (**F**) Schematic of Ire1–GFP and Ire1–mCherry with the fluorescent modules placed in the juxtamembrane region of the cytosolic linker. (**G**) Localization of Ire1–mCherry WT or IF$^C$ mutants, as well as SpR$^{U1A}$ mRNA decorated with U1A–GFP, after induction of ER-stress with DTT. (**E**, **G**) ER-stress was induced with 10 mM DTT for 45 min; imaging was performed in *ire1Δ* cells, complemented with Ire1 imaging constructs, as described (***Aragón et al., 2009***). Scale bars represent 5 μm. (**H**) Immunoblot of hemagglutinin (HA)-tagged Ire1 protein from lysates from strains as in panel (**D**) and (**G**). A sample from a strain that overexpressed HA-tagged Ire1 from a 2 μ plasmid served as a positive reference. Ire1 is denoted by an arrowhead. A background band, denoted by an asterisk (*), conveniently serves as a loading reference. (**I**) Bar diagrams depict the percentage of Ire1 signal in foci (red bars) and the co-localization index expressed in arbitrary units (yellow bars), as described (***Aragón et al., 2009***), for SpR$^{U1A}$ mRNA recruitment to foci of Ire1 variants shown in (**G**); mean and s.e.m. are shown, n = 5–8. There is no statistical significance in *a Student's t-test* of differences in foci formation and mRNA recruitments as compared with wild-type.

The following source data is available for figure 1:

**Source data 1**. (**A**) Source data for *Figure 1D* and *Figure 1H*.

is compatible with the size of Ire1 foci observed by fluorescence microscopy (***Aragón et al., 2009***) and is sterically feasible despite the twists of the filaments on either side of the planar membrane, owing to the flexibility and length (>100 Å) of the linker domains on either side of the membrane, which can relieve the strain. Alternatively and not mutually exclusive, Ire1 clusters may be dynamic, such that constant rearrangements of the Ire1 molecules in clusters sustain transient intermittent oligomerization events on either side of the membrane.

We previously have shown that Ire1 oligomerization allows selective recruitment of unspliced *HAC1* mRNA to Ire1 clusters by virtue of a bipartite element in *HAC1*'s 3′ untranslated region (UTR) (***Aragón et al., 2009***), which we named the 3′ BE. The 3′ BE is effective in targeting mRNA to Ire1 clusters as long as they are translationally repressed (***Aragón et al., 2009***). Stalling of *HAC1* mRNA translation is afforded through base pairing between the intron and the 5′ UTR (***Rüegsegger et al., 2001***). Moreover, the in vitro endonuclease activity of Ire1 kinase/RNase domains is highly cooperative, indicating that oligomerization rather than dimerization leads to RNase activation (***Korennykh et al., 2009***). Intriguingly, the capacity of the kinase/RNase domains to oligomerize in vitro depends on a short stretch of the cytosolic linker that extends N-terminally from the kinase domain (***Korennykh et al., 2009***).

In this study, we report that although oligomeric assembly of kinase/RNase domains is essential for activation of Ire1's in vivo mRNA processing capacity, it is not required for driving formation and maintenance of Ire1 clusters or for recruitment of its mRNA substrate. Instead, we discovered that a conserved positively charged element in Ire1's cytosolic linker mediates mRNA docking onto Ire1 clusters. Primary docking of the mRNA to this site is required for subsequent processing of the mRNA by Ire1's RNase domain. The staged way in which *HAC1* mRNA is channeled to become subject to Ire1's endonuclease activity enhances efficiency and selectivity in the process and, thus, fidelity in UPR signaling.

## Results

### Oligomeric assembly of the kinase/RNase domains is essential for Ire1 function but not for its clustering or for *HAC1* mRNA recruitment

Efficiency of *HAC1* mRNA splicing (***Figure 1C***) depends on clustering of ER-lumenal domains in vivo (***Kimata et al., 2007***; ***Aragón et al., 2009***) and of cytosolic domains in vitro (***Korennykh et al., 2009***). To assess the contribution of cytosolic oligomerization events to Ire1 function in vivo, we analyzed mutations in the interfaces that govern Ire1 kinase/RNase oligomeric assembly by complementing *ire1Δ* yeast with centromeric plasmids. Driven by its autologous promoter, expression of (wild-type or mutant) Ire1 from these plasmids is at near-endogenous levels (***Aragón et al., 2009***). Disruption of IF1$^C$ abolished RNase function (***Figure 1D***) as monitored by the loss of expression of a fluorescent reporter protein (SpR; ***Figure 1C***), whose levels report on Ire1 RNase activity (***Pincus et al., 2010***). Consequently, growth under ER-stress conditions was impaired (***Figure 1D***), as previously reported (***Lee et al., 2008***). Disruption of IF2$^C$ likewise disrupted RNase function and survival under ER-stress (***Figure 1D***), consistent with in vitro analyses (***Korennykh et al., 2009***). Mutations in IF3$^C$ led to a

milder phenotype, sustaining intermediate levels of splicing and growth (*Figure 1D*), similar to the lumenal interface mutants that are shown for comparison (*Aragón et al., 2009*; *Figure 1D*). As expected, mutations in Ire1 did not affect growth under non-ER-stress conditions since growth of *ire1Δ* yeast is then also unaffected (*Aragón et al., 2009*).

Under control conditions, Ire1 distributed diffusely throughout the ER but clustered into discrete foci upon ER-stress as visualized with fluorescently tagged Ire1–GFP (*Aragón et al., 2009*; *Figure 1E,F*). *HAC1* mRNA is recruited to Ire1 foci under ER-stress via the 3′ BE targeting signal in the mRNA (*Aragón et al., 2009*; *Figure 1C*), which can be visualized in cells expressing *HAC1* splicing reporter mRNA containing U1A binding sites (SpR$^{U1A}$, *Figure 1C*) and GFP-tagged U1A RNA-binding protein (*Brodsky and Silver, 2000*; *Aragón et al., 2009*). Disruption of lumenal interfaces interfered with Ire1 clustering (*Kimata et al., 2007*; *Aragón et al., 2009*; *Figure 1E*) and, consequently, mRNA recruitment (*Aragón et al., 2009*).

Disruption of cytosolic interfaces also compromised splicing activity of Ire1 and, hence, growth under ER-stress conditions (*Figure 1D*), but foci formation was unaffected (*Figure 1G*). Moreover, expression levels of the IF$^C$ mutants were comparable to wild-type (*Figure 1H*), and the percentage of Ire1 in foci as well as the extent of co-localization of mRNA at those foci, as determined by a customized MatLab script (*Aragón et al., 2009*), was similar for wild-type and IF$^C$ mutants (*Figure 1G,I*). Thus, all three cytosolic oligomeric interfaces are key for Ire1 function in vivo but not for Ire1 stability or its capacity to cluster and recruit *HAC1* mRNA.

## The kinase and RNase domains are dispensable for Ire1 clustering and mRNA recruitment

Tampering with the oligomeric assembly of the cytosolic domains of Ire1 gravely affected Ire1's endonuclease activity. To explore directly whether enzymatic activity of Ire1 is necessary for foci formation, we extended our assays using site-specific mutations that selectively disrupt Ire1's kinase and RNase activities ('KD' and 'RD' for kinase- and RNase-deficient, respectively). In line with previous results (*Papa et al., 2003*; *Korennykh et al., 2011*), splicing and survival under ER-stress were impaired in *ire1(KD)* and abolished in *ire1(RD)* mutant cells, while mutant Ire1 expression levels were similar to wild-type (*Figure 2A*). Yet, Ire1 foci formation and mRNA recruitment were unimpeded in either mutant (*Figure 2B,F*), indicating that neither of the enzymatic activity is required for Ire1 to recruit its mRNA substrate.

Strikingly, even removal of the entire RNase domain, either alone ('ΔR') or together with the kinase domain ('ΔKR'), left Ire1 foci formation and mRNA recruitment unaffected (*Figure 2C,F*, *Figure 2—source data 1*). These results show that oligomeric assembly of the kinase–RNase domain is dispensable for the Ire1 clustering. Moreover, these findings indicate that Ire1 must harbor an mRNA docking site within the linker that tethers the kinase/RNase to the transmembrane region because it is the only remaining cytosolic portion in the *ire1(ΔKR)* mutant. A heterologous mRNA, SL-*PGK*-3′ *hac1*$^{U1A}$—which contains the 3′ BE of *HAC1* mRNA, but lacks the *HAC1* mRNA intron and splice sites, and a small stem-loop in its 5′ UTR to repress its translation (*Aragón et al., 2009*; and *Figure 2D*)—was also efficiently recruited to foci in *ire1(ΔKR)* mutant cells (*Figure 2E,F*, *Figure 2—source data 1*). These results were surprising, since mRNA recruitment to Ire1 serves to engage the splice sites in the *HAC1* mRNA with Ire1's endonuclease domain for cleavage, yet neither the endonuclease domain nor the splice sites (or their context) are required for mRNA docking onto Ire1 clusters. Instead, our findings indicate that the core elements sufficient for the recruitment of *HAC1* mRNA to and docking of the mRNA onto Ire1 clusters are contained in the 3′ BE of the mRNA and in Ire1's cytosolic linker domain.

## The cytosolic linker of Ire1 harbors a conserved positively charged motif that facilitates mRNA docking

Ire1 is the only ER-stress sensor that is conserved in all eukaryotes. The kinase/RNase domains and the core lumenal ER-stress sensing domain are conserved, but other domains, including the cytosolic linker domain, show negligible sequence conservation (*Figure 3A*). The linker greatly varies in length between species but consistently harbors an unusually high number of basic and acidic residues. In fungal species, a short basic sequence stretch (henceforth referred to as '[+]-box') displays recognizable homology. In particular, sequence alignment reveals strict conservation of one lysine and two arginine residues as well as three glycine residues that intersperse the basic residues. The [+]-box is flanked on either one or both sides by acidic sequences.

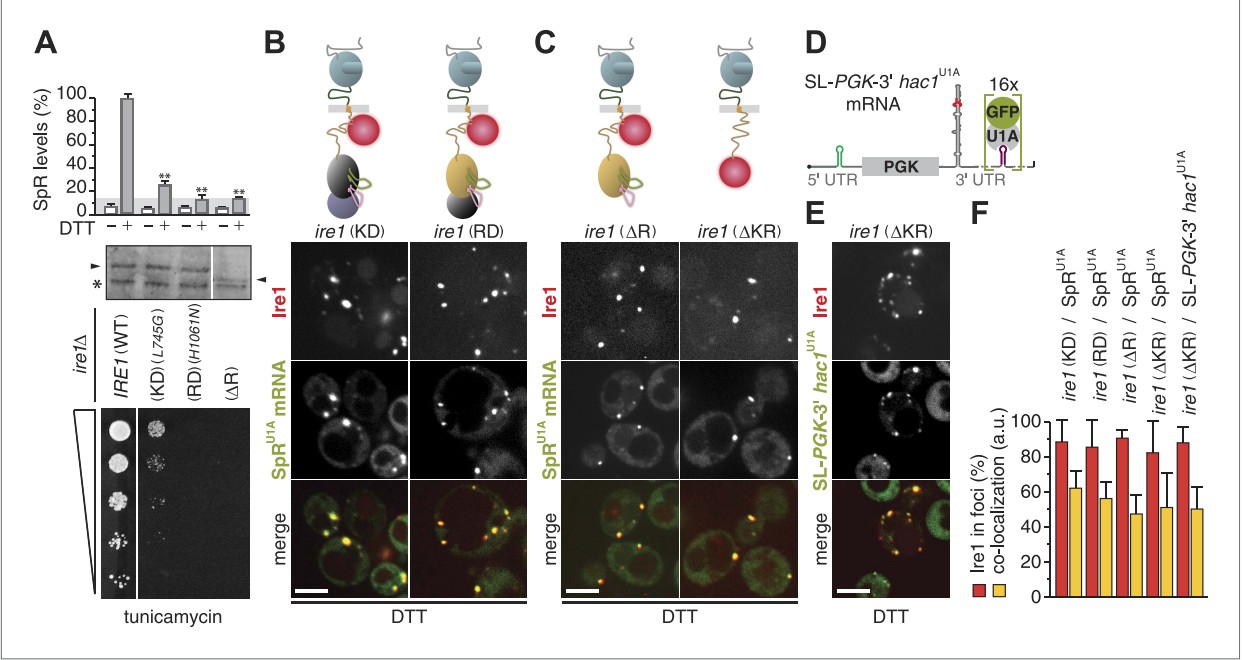

**Figure 2**. The kinase and RNase domains of Ire1 are dispensable for foci formation and mRNA recruitment. (**A**) Splicing reporter assay before or after ER-stress induction with 2 mM DTT for 2 hr (top), Western blot of Ire1 (middle), and viability assay under ER-stress conditions (0.2 µg ml⁻¹ tunicamycin; bottom) were performed in *ire1Δ* yeast containing a genomic copy of the SpR, complemented with wild-type (WT), kinase dead (KD), RNase dead (RD), and RNase truncation (ΔR) mutant alleles of ire1. Maximal (100%) and background level (14%, light gray bar) fluorescence are set as in **Figure 1A**. Mean and s.d. are shown (n = 2). Statistical significance in *a Student's t-test* of differences in splicing levels as compared with wild-type is indicated (**p ≤ 0.01). The arrowheads denote (mutant or truncated) Ire1 protein and the asterisk a background band on the immunoblot as in **Figure 1H**. (**B**, **C**) Top: schematic of the mCherry-tagged versions of the same ire1 mutants as in (**A**) as well as a kinase/RNase truncation (ΔKR) mutant, color-coded as in **Figure 1A**, except defective domains are black. (**D**) Schematic of a chimeric mRNA, SL-*PGK1*-3' *hac1*[U1A], which is *PGK1*[U1A], bearing in its 3′ UTR the stem-loop structure with the 3′ BE of the *HAC1* mRNA and in its 5′ UTR a small stem-loop (green) that confers translational repression (**Aragón et al., 2009**). (**B**, **C**, **E**) Localization of Ire1–mCherry and of U1A–GFP decorating either SpR[U1A] (**B**, **C**) or SL-*PGK1*-3' *hac1*[U1A] (**E**) mRNA. ER-stress was induced with 10 mM DTT for 45 min; imaging was performed of *ire1Δ* cells, complemented with Ire1 imaging constructs, as depicted. Scale bars represent 5 µm. (**F**) Bar diagrams depict the percentage of Ire1 signal in foci (red bars) and the co-localization index for mRNA recruitment into foci of Ire1 variants shown in **B**, **C**, and **E** (mean and s.e.m., n = 5–10). There is no statistical significance in *a Student's t-test* of differences in foci formation and mRNA recruitments as compared with wild-type.

The following source data is available for figure 2:

**Source data 1**. (**A**) Source data for **Figure 2A** and **Figure 2F**.

All fungal species with a [+]-box containing Ire1 linker have a conserved 3′ BE in their *HAC1* mRNAs (**Aragón et al., 2009**). The only fungal species we found with a markedly divergent basic motif is *Schizosaccharomyces pombe*, which lacks a *HAC1* gene altogether (**Kimmig et al., 2012**). In both Ire1 paralogs of *Arabidopsis thaliana*, the linkers harbor a basic motif that diverges from the fungal [+]-box (**Figure 3A**), but that motif is conserved among plants (not shown). Conversely, animal species lack any such recognizable motif (**Figure 3A**).

Indicative of an important role for the [+]-box in UPR signaling is that it adorns the short linker extension, which facilitated oligomerization and markedly enhanced endonuclease activity of recombinant kinase/RNase domains in vitro (**Korennykh et al., 2009**). To analyze the role for the [+]-box in Ire1 function in vivo, we first truncated Ire1 further such that the [+]-box was deleted (**Figure 3B**). Since we could not obtain a centromeric plasmid of this construct, as it was toxic for *Escherichia coli*, we created *ire1Δ* strains with a genomic copy of the *ire1ΔKR* or *ire1ΔKR/Δ[+]-box* mCherry-tagged transgenes in the *LEU2* locus. As expected, Ire1 clustering and mRNA recruitment were still at wild-type levels for *ire1ΔKR* when genomically integrated (**Figure 3B**), similar to what we observed when *ire1ΔKR* was expressed from a centromeric plasmid (**Figure 2C,F**). Further removal of the [+]-box did not affect Ire1 clustering but markedly reduced mRNA recruitment (**Figure 3B**). This finding suggests

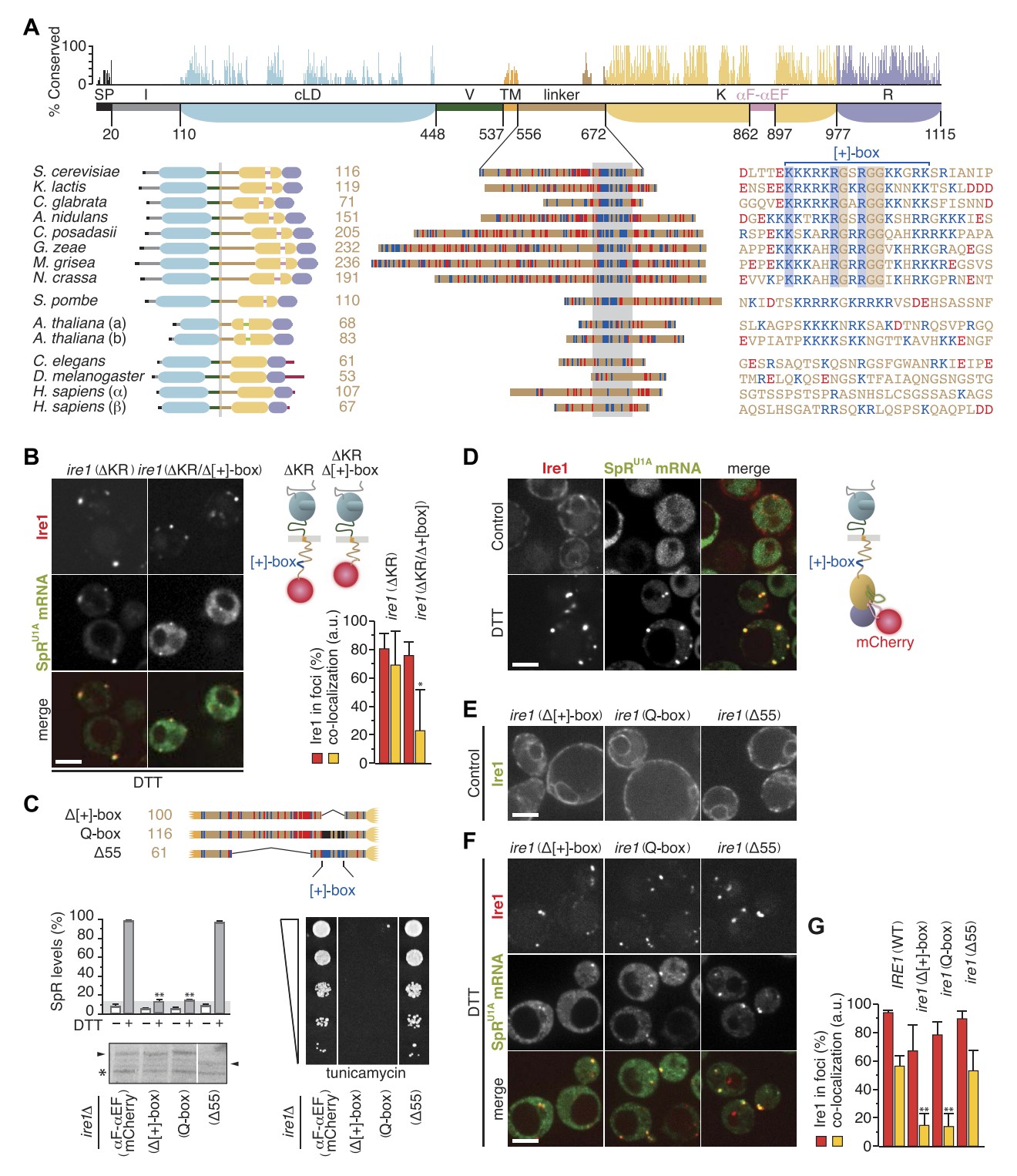

**Figure 3**. The cytosolic linker of Ire1 harbors a positively charged motif that is key for mRNA recruitment and splicing. (**A**) Conservation of Ire1. Top: mapped onto a schematic of Ire1 domains bordered by residues of which the number is denoted, bar diagrams display relative conservation of the *Saccharomyces cerevisiae* Ire1 protein sequence to homologs (lower, left) from other fungal species *Kluyveromyces lactis*, *Candida glabrata*, *Aspergillus nidulans*, *Coccidioides posadasii*, *Gibberella zeae*, *Magnaporthe grisea*, *Neurospora crassa*, and *Schizosaccharomyces pombe*, as well as from the animals *Caenorhabditis elegans* and *Drosophila melanogaster*, the two paralogues from the plant *Arabidopsis thaliana* (a and b) and from *Homo sapiens* (α and β). Domains are color-coded as in *Figure 1A*, except signal peptides (SP) are black; light green represents a loop inserted into the kinase domain

*Figure 3. Continued on next page*

*Figure 3. Continued*

of the *A. thaliana* Ire1s and crimson denotes C-terminal extensions in animal Ire1s. Expanded view (middle) of the linker domains that are aligned based on the stretch (gray box) for which the sequence alignment is shown on the right. Strictly conserved residues among fungal species except *S. pombe* are boxed. (**A**) lower right, (**C**) top, Basic (arginine and lysine) residues are shown in blue and acidic (aspartate and glutamate) residues in red. Glutamines replacing arginines and lysines in the Q-box mutant are black. (**A**) lower right, (**B**, **C**) Position of the [+]-box is indicated. (**B**, **D–F**) Localization of Ire1–GFP or Ire1–mCherry and of U1A–GFP decorated SpRU1A mRNA. Imaging was performed in *ire1Δ* yeast complemented with a genomic copy of C-terminally mCherry-tagged (ΔKR) or (ΔKR/Δ[+]-box) ire1 mutant alleles, as schematically shown (**B** top right), or with plasmids encoding IRE1 wild-type (**C**) or ire1 linker mutants (**E**, **F**), with the fluorescent modules GFP (**E**) and mCherry (**C**, **F**) placed in the αF–αEF loop, as schematically shown (**D** right), before (**D** upper panels, **E**, control) and after (**B**, **D** lower panels, **F**, DTT) induction of ER-stress with 10 mM DTT for 45 min. Scale bars represent 5 µm. Bar diagrams depict the percentage of Ire1 signal in foci (red bars) and the co-localization index for mRNA recruitment into foci of Ire1 variants (mean and s.e.m., n = 5–10) (**B**, bottom right, **G**). Statistical significance in *a Student's t-test* of differences in foci formation and mRNA recruitments as compared with wild-type is indicated (*p ≤ 0.05; **p ≤ 0.01). (**C**) Schematic of linker domains with mutations or truncations as in (**A**) (top). Splicing reporter assay before or after ER-stress induction with 2 mM DTT for 2 hr (left, middle), Western blot of Ire1 (left, bottom), and viability assay under ER-stress conditions (0.2 µg ml⁻¹ tunicamycin; right, bottom). Assays were performed in *ire1Δ* yeast containing a genomic copy of the SpR, complemented with either IRE1 wild-type or ire1 linker mutants with mCherry in the αF–αEF loop. Mean and s.d. are shown (n = 2). Maximal (100%) and background level (14%, light gray bar) fluorescence are set as in *Figure 1A*. Statistical significance in *a Student's t-test* of differences in splicing levels as compared with wild-type is indicated (**p ≤ 0.01). The arrowheads denote (mutant or truncated) Ire1 protein and the asterisk a background band on the immunoblot as in *Figure 1H*.

The following source data is available for figure 3:

**Source data 1**. (**A**) Source data for *Figure 3B*, *Figure 3C* and *Figure 3G*.

that the [+]-box is key for the docking of mRNA onto Ire1 clusters. Moreover, given that the extent of clustering of the *ire1 (ΔKR/Δ[+]-box)* mutant was similar to wild-type, we conclude that Ire1 clustering is driven by the lumenal domain alone: neither cytosolic oligomeric assembly (as facilitated by the kinase/RNase domains) nor mRNA docking (as facilitated by the [+]-box) is required for Ire1 foci formation.

Next, we set out to explore the role of the [+]-box in the context of the full-length Ire1 using the mutants and truncations depicted in *Figure 3C*. The experiments presented so far employed Ire1 variants bearing GFP or mCherry modules in the linker (*Aragón et al., 2009*; *Pincus et al., 2010*; *Rubio et al., 2011*; *Figure 1F*; *Figure 2B,C*). To avoid interference with mutations in the linker, we relocated the fluorescent modules to the αF–αEF loop in the Ire1 kinase domain (*Figure 3D*). By contrast to the activation loop (*Figure 1A*), which becomes hyperphosphorylated due to Ire1's kinase activity, the αF–αEF loop is poorly conserved (*Figure 3A*) and dispensable for Ire1 activity in vitro (*Korennykh et al., 2009*). Accordingly, despite the insertion of a fluorescent protein module into the αF–αEF loop, Ire1 still faithfully formed foci and recruited SpRU1A mRNA under ER-stress conditions (*Figure 3D,G*) and was indistinguishable from untagged wild-type Ire1 in splicing efficiency and growth under ER-stress conditions (*Figure 3C*).

Removal of the [+]-box abolished splicing and growth under ER-stress (*Goffin et al., 2006*; *Figure 3C*, 'Δ[+]-box'), as did replacement of lysines and arginines within the [+]-box with glutamines (*Figure 3C*, 'Q-box'). These mutants were expressed at similar levels as the wild-type (*Figure 3C*). However, extensive truncation of the rest of the linker ('Δ55')—to a length comparable to the shortest among all Ire1 homologs (61 amino acids; *Figure 3A*)—had no effect on growth (*Figure 3C*). The [+]-box was shown previously to act as a nuclear import signal when inserted into heterologous proteins (*Goffin et al., 2006*). For Ire1, however, such a role is unlikely, since tampering with the [+]-box had no effect on the distribution throughout the ER on either of non-clustered Ire1 in the absence (*Figure 3E*) or of Ire1 foci in the presence (*Figure 3F*) of ER-stress. Conversely, removal or mutagenesis of the [+]-box did affect the co-localization of the SpRU1A mRNA to Ire1 foci (*Figure 3F,G*). Shortening of the linker, while leaving the [+]-box intact, did not impact on co-localization of the mRNA with Ire1 clusters (*Figure 3F,G*). Thus, the splicing and growth defects (*Figure 3D*) correlated with impairments in mRNA recruitment (*Figure 3F*). We pose that the [+]-box is a key element for Ire1 function in vivo, because it facilitates docking of the mRNA onto Ire1 clusters.

## Three arginine residues are key for mRNA docking at the [+]-box

To assess which of the positively charged residues within the [+]-box are important for mRNA docking, we next replaced each individual lysine or arginine with threonine. Splicing and growth under ER-stress were abrogated in mutants of either conserved arginine, R647 or R650 (*Figure 4A*). A third,

less-conserved arginine, R645, appeared almost equally important for Ire1 function (***Figure 4A***). Replacement of all non-basic residues (including the three conserved glycines) in the [+]-box with arginines or lysines (***Figure 4A***, 'all [+] mutant') disrupted splicing and survival upon ER-stress almost to the level of the mutants of the conserved arginines. This observation suggests that a positively charged cluster alone is not sufficient but that the glycines afford proper display (spacing or three-dimensional positioning) of the crucial arginine side chains.

Mutation of no other basic residue, including the single conserved lysine (K642T), markedly disturbed splicing or growth under ER-stress (***Figure 4A***). Accordingly, mRNA recruitment was unaffected in any of these mutants (not shown). By contrast, mRNA recruitment was impaired for each mutant of the three key arginines, as well as for the 'all [+]' mutant (***Figure 4B,C***), though they were expressed at similar levels to wild-type (***Figure 4D***). The defect in mRNA recruitment for the R647T and R650T

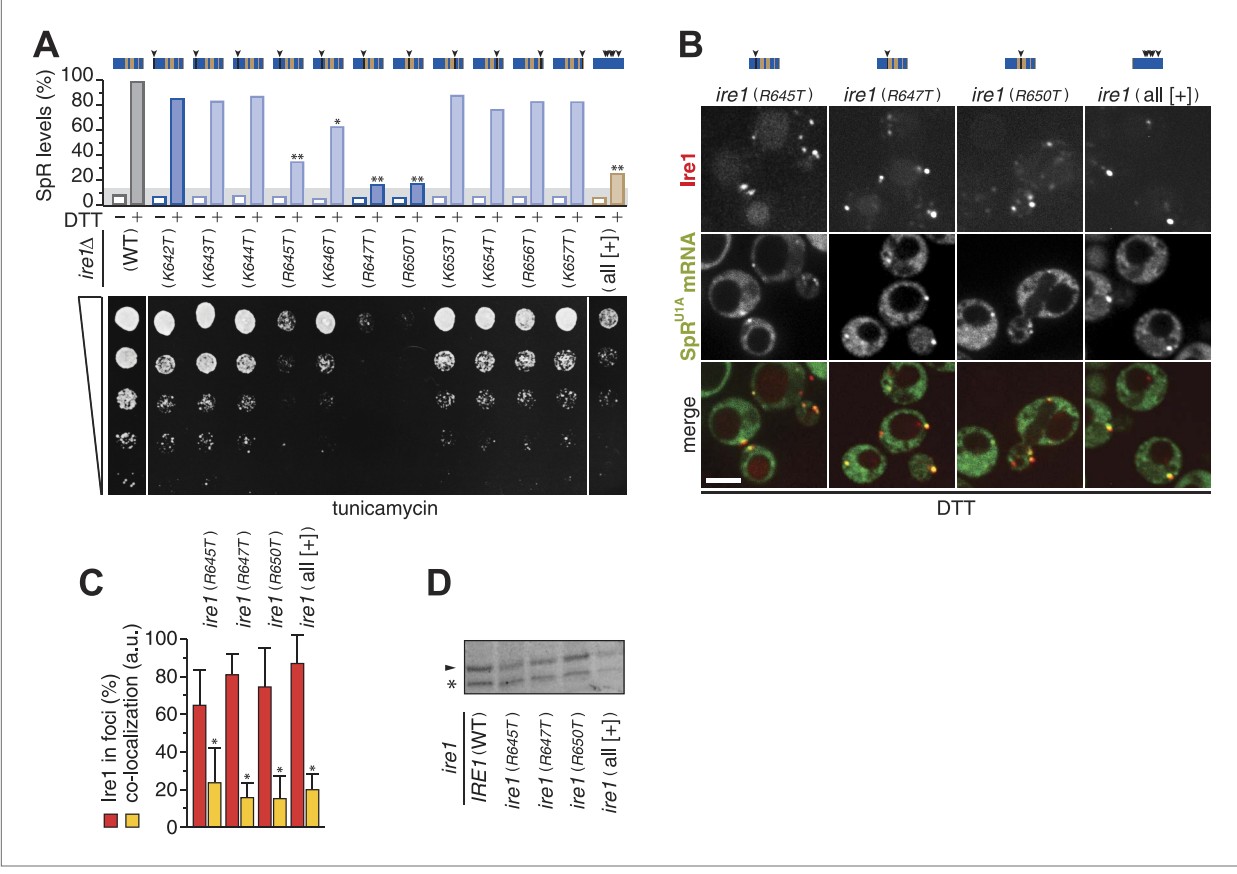

**Figure 4**. Three arginines in Ire1's linker are essential for mRNA docking. (**A**) Splicing reporter assay before or after ER-stress induction with 2 mM DTT for 2 hr (top) and viability assay under ER-stress conditions (0.2 µg ml$^{-1}$ tunicamycin; bottom). Assays were performed in *ire1Δ* yeast containing a genomic copy of the SpR, complemented with ire1 mutants having within the [+]-box a single arginine or lysine replaced with a threonine, as indicated, or all non-positively charged residues replaced with arginines or lysines (KKKRKRKKKRKRKKKRRK; all [+]). Bar diagrams reporting on mutants of positively charged residues are blue with conserved residues in a darker shade; bar diagrams of the all [+] box mutant in brown. Statistical significance in *a Student's t-test* of differences in splicing levels as compared with wild-type is indicated (*p ≤ 0.05; **p ≤ 0.01). Maximal (100%) and background level (14%, light gray bar) fluorescence are set as in ***Figure 1D***. (**B**) Localization of Ire1–mCherry and U1A–GFP decorated SpR$^{U1A}$ mRNA. Imaging was performed in *ire1Δ* yeast complemented with ire1 linker mutants having the mCherry module in the αF–αEF loop. Scale bar represents 5 µm. (**A**, **B**) Schematics of [+]-box variants are color-coded as in ***Figure 3A*** with arrow heads denoting the position of point mutations in black. (**C**) Co-localization index for mRNA recruitment into foci of Ire1 variants shown in **B** (mean and s.e.m., n = 5–10). Statistical significance in *a Student's t-test* of differences in foci formation and mRNA recruitments as compared with wild-type is indicated (*p ≤ 0.05). (**D**) Immunoblot as in ***Figure 1H*** of Ire1 of lysates from strains in panel (**B**); the arrowhead denotes Ire1 protein and the asterisk a background band as in ***Figure 1H***.

The following source data is available for figure 4:

**Source data 1**. (**A**) Source data for ***Figure 4A*** and ***Figure 4C***.

mutants in particular was as strong as the impairment caused by the deletion of the entire [+]-box (*Figure 3F*). We conclude that three arginine residues, R645, R647, and R650, in a proper spatial arrangement are key for the [+]-box to sustain mRNA docking.

## *HAC1* mRNA targeting to, docking onto, and splicing by Ire1 clusters are separate steps in UPR signaling

Removal of the 3′ BE targeting element from the mRNA resulted in a complete loss of co-localization with Ire1 foci (*Aragón et al., 2009*; *Figure 5A*). In contrast, the loss of a functional [+]-box did not completely eliminate co-localization of the mRNA with Ire1 foci; rather, co-localization levels were reduced two- to threefold compared to the wild-type (*Figure 3B,F*; *Figure 4B,C*). Apparently, targeting of the mRNA via the 3′ BE permits residual co-localization with the [+]-box mutants. Such residual co-localization was evident even for the *ire1(ΔKR/Δ[+]-box)* mutant (*Figure 3B*), implying that 3′ BE-mediated targeting occurs irrespective of any cytosolic portion of Ire1.

Paradoxically, the residual co-localization of the mRNA with Ire1 clusters that have a defective [+]-box (*Figure 3F,4B,C*) was not sufficient to sustain splicing (*Figure 3C,4A*), despite the fact that Ire1's kinase and RNase domains were intact. Since the [+]-box is unlikely to contribute to the core enzymatic function of Ire1, as it is not conserved in all eukaryotes, we reasoned that the docking mechanism itself is important to initiate splicing activity. To test this idea, we created a synthetic bypass for [+]-box mediated docking by inserting the U1A RNA binding module into the αF–αEF loop of the *ire1(Δ[+]-box)* mutant. We reasoned that the insertion of the U1A module would allow direct interaction of the Ire1–U1A fusion protein with *HAC1*[U1A] mRNA via the U1A-binding hairpins that we had introduced into the 3′ UTR (*Figure 5B*), which we thus far employed in the mRNAs for visualization purposes (*Figure 1C*). Strikingly, the U1A module considerably restored splicing and growth under ER-stress conditions of *ire1(Δ[+]-box)* mutant complemented with *HAC1*[U1A] mRNA (*Figure 5C*). Thus, docking of the mRNA per se to a site distinct from the kinase/RNase domain is key for the mRNA to be processed by Ire1. The [+]-box is the module that affords such docking for Ire1, but it can be substituted by an alternative means that provides an mRNA docking platform.

We have shown before that the removal of the 3′ BE from *HAC1* leads to a substantial reduction of splicing efficiency and growth under ER-stress conditions (*Aragon et al., 2009*). Commensurate with that loss, splicing and growth were restored to a lesser degree when the *ire1Δ[+]-box*-U1A mutant was complemented with *HAC1* Δ3′ BE [U1A] mRNA than when the 3′ BE was present (*Figure 5C*). This finding confirms that 3′ BE-mediated mRNA targeting is an event separate from mRNA docking (whether mediated by the [+]-box or the U1A 'bypass'). Taken together, these results indicate that the engagement of *HAC1* mRNA with Ire1 is a step-wise process. The 3′ BE directs translationally repressed *HAC1* mRNA to sites where Ire1 clusters, which is driven solely by oligomeric assembly of its LD. Docking of *HAC1* mRNA onto the [+]-box then is a pre-requisite for productively engaging *HAC1* mRNA with Ire1's endoribonuclease, whereupon *HAC1* mRNA is cleaved.

## Discussion

Our results define the [+]-box within the cytosolic linker domain of Ire1 as a critical mRNA docking site. Docking of *HAC1* mRNA at the [+]-box is a pre-requisite for Ire1 activity in vivo, in agreement with our previous finding that the [+]-box-containing linker enhances the RNase activity of Ire1 by more than 100-fold in vitro (*Korennykh et al., 2009*). These results suggest a previously unrecognized step in which *HAC1* mRNA participates in Ire1 activation to promote its own cleavage and ascertain high specificity and are summarized in a speculative model in *Figure 6*

### Step 1 (Ire1-clustering)

Oligomeric assembly on both sides of the ER membrane is required for Ire1 function. Unfolded protein binding to the Ire1-LDs leads to Ire1 oligomerization in foci. As we show here, foci formation does not require participation of Ire1's cytoplasmic domains. It coincides with the loss of BiP binding to the Ire1 lumenal domain (not shown) and utilizes interfaces IF1[L] and IF2[L]. LD-driven clustering concentrates Ire1's kinase/RNase modules on the cytosolic side of the ER membrane and enables formation of the oligomerization interfaces IF1[C], IF2[C], and IF3[C] of the cytoplasmic domains that organize and activate the RNase domains.

### Step 2 (mRNA-targeting)

*HAC1* mRNA is targeted to the foci via its 3′ BE, as long as the mRNA is translationally repressed (*Aragón et al., 2009*). The molecular machinery for this process remains unknown. Targeting of mRNAs

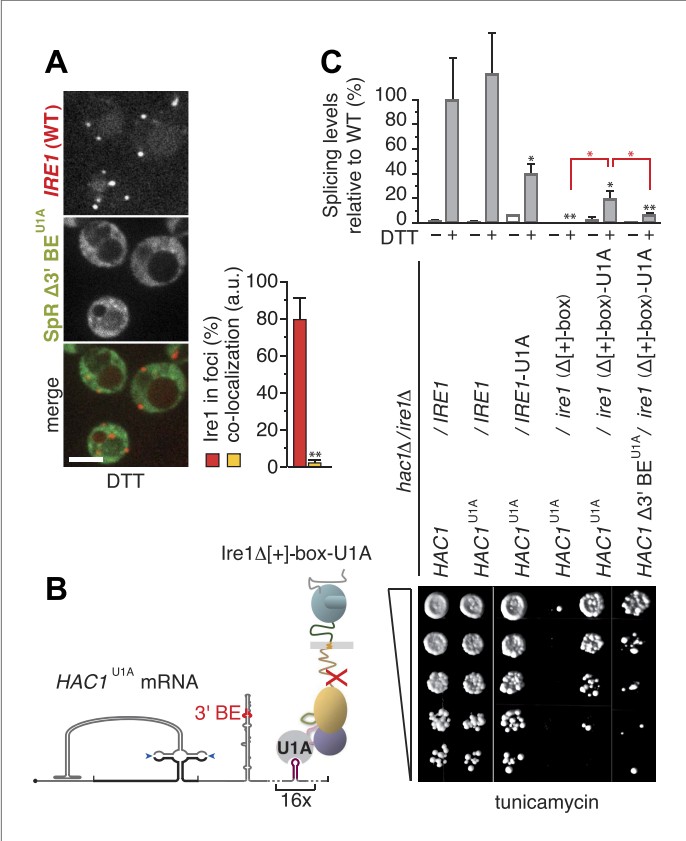

**Figure 5**. Step-wise targeting and docking of mRNA are pre-requisite for activating Ire1. (**A**) Localization of mCherry-tagged Ire1 and SpR Δ3' BE^U1A mRNA decorated with U1A–GFP. ER-stress was induced with 10 mM DTT for 45 min; imaging was performed in *ire1Δ* cells, complemented with wild-type *IRE1* having the mCherry module in the αF–αEF loop (left). Scale bar represents 5 µm. Co-localization index for mRNA recruitment into foci (mean and s.e.m., n = 10) (right). Statistical significance of the difference in mRNA recruitment of SpR Δ3' BE^U1A as compared with SpR^U1A to Ire1–mCherry clusters (*Figure 3D,G*) was tested using *a Student's t-test* (**p ≤ 0.01). (**B**) Schematic of mRNA docking 'bypass'. The U1A module placed in the αF–αEF loop of Δ[+]-box mutant ire1 facilitates binding of *HAC1*^U1A mRNA via its U1A motifs. (**C**) Splicing was measured by quantitative RT-PCR before or after ER-stress induction with 2 mM DTT for 30 min (top) and viability assay under ER-stress conditions (0.2 µg ml⁻¹ tunicamycin) of *hac1Δ/ire1Δ* yeast complemented with centromeric plasmids bearing wild-type IRE1 or Δ[+]-box mutant ire1 either untagged or tagged with the U1A module in the αF–αEF loop, as well as with centromeric plasmids bearing wild-type *HAC1*, *HAC1*^U1A, or *HAC1* Δ3' BE^U1A. For display of RT-PCR results, the signal for *hac1Δ/ire1Δ* yeast complemented with wild-type *HAC1* and IRE1 under ER-stress conditions was set at 100%; mean and s.d. (n = 2) are shown. Statistical significance in *a Student's t-test* of differences in splicing levels as compared with wild-type is indicated in black and of differences in splicing levels compared with the 'bypass' (*HAC1*^U1A + *ire1* Δ[+]-box-U1A) is indicated in red (*p ≤ 0.05; **p ≤ 0.01).

The following source data is available for figure 5:

**Source data 1**. (**A**) Source data for *Figure 5A* and *Figure 5C*.

bearing a 3' BE can occur to Ire1 foci lacking Ire1's cytoplasmic domains, indicating that a putative mRNA targeting receptor(s) may exist (not shown in *Figure 6*), which senses that Ire1-LDs are clustered.

## Step 3 (mRNA-docking)

Concentration of RNAs by recruitment to foci allows docking onto the [+]-boxes in the Ire1 cytosolic linker. As we show here, a synthetic bypass can substitute for the [+]-box mediated docking. Docking may fortify oligomeric assembly of the cytosolic Ire1 domains by tethering the [+]-boxes of several clustered Ire1 monomers. Accordingly, the [+]-box may serve a role akin to the arginine-rich

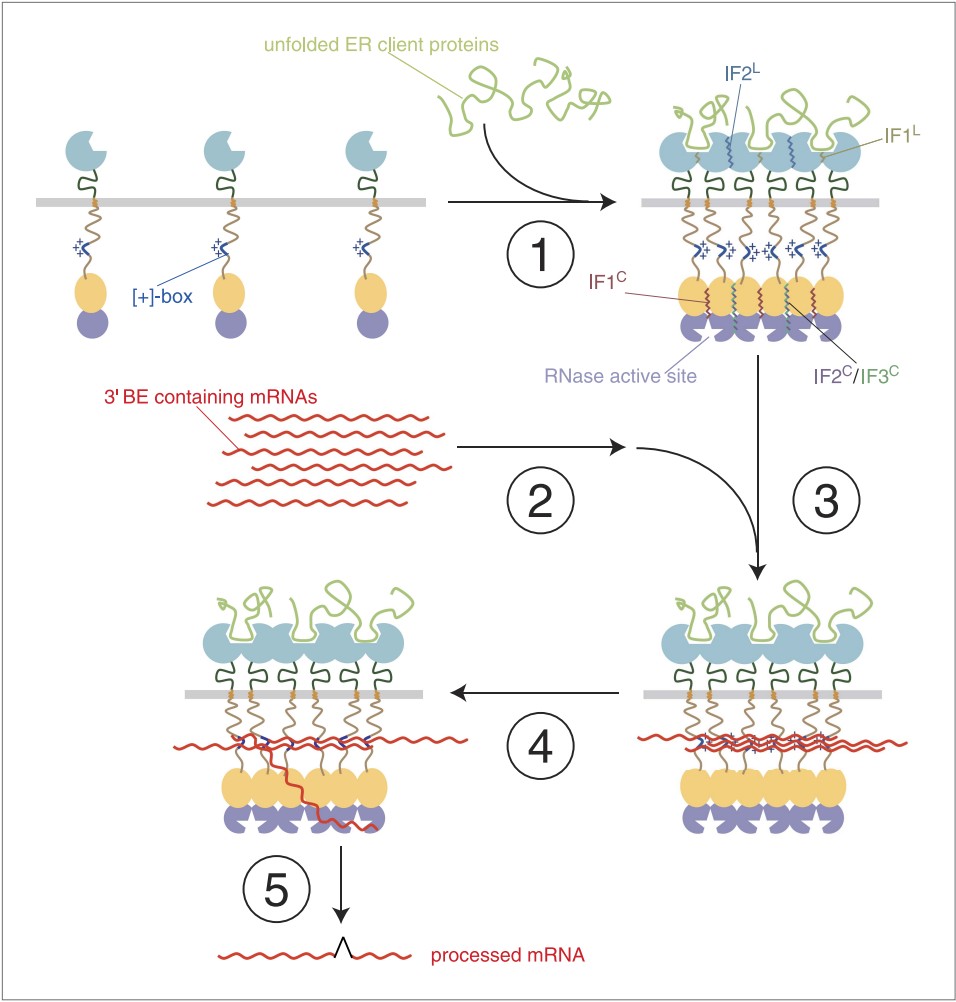

**Figure 6**. A model of step-wise activation of the Ire1/*HAC1* mRNA signaling relay. The steps are described in the Discussion. Color-coding is as in *Figure 1*, except that unfolded proteins are depicted in light green and mRNA in red. Ire1 domain interfaces are indicated by wiggly lines.

domain of HIV1 Rev, which promotes both RNA binding and protein oligomerization (*Zapp et al., 1991*; *Daugherty et al., 2010*).

## Step 4 (priming)
*HAC1* mRNA is either repositioned from the [+]-box-docked pool (as depicted) or newly targeted *HAC1* mRNA molecules are recruited and engaged with the active site of the RNase domain.

## Step 5 (RNA cleavage and ligation)
*HAC1* mRNA is cleaved and the severed exons are ligated to produce the spliced mRNA product of the reaction.

Our findings identify three arginine residues within the [+]-box that are crucial for mRNA docking and splicing. A key contribution of arginine residues is common to RNA binding proteins (*Lazinski et al., 1989*; *Tan and Frankel, 1995*), suggesting that mRNA binding to the [+]-box is direct. In particular, the [+]-box encompasses a conserved arginine-glycine-glycine (RGG) motif, which in the form of repeats adorns several RNA binding protein families, including snRNPs, hnRNPs, and snoRNPs (*Godin and Varani, 2007*). Via combined low-affinity interactions, RGG boxes synergize in the assembly of RNA–protein complexes (*Godin and Varani, 2007*). Along these lines it is plausible that when Ire1 is monomeric, the [+]-boxes of individual Ire1 molecules may not sustain RNA binding, which can explain

the lack of mRNA recruitment under non-ER-stress conditions (*Aragón et al., 2009*; *Figure 3D*). Clustering of [+]-boxes, resulting from Ire1 LD-driven oligomerization, would generate the required synergistic avidity, such that only clustered [+]-boxes form a docking platform for the mRNA.

We found that residual targeting of *HAC1* mRNA occurs even in the absence of recognizable features conserved among Ire1 proteins at the cytosolic side of the membrane (*ire1ΔKR/Δ[+]-box*). Moreover, even when mRNA docking was facilitated by the U1A-mediated 'bypass', the 3′ BE still contributed to *HAC1* mRNA splicing efficiency. These data indicate that 3′ BE-mediated *HAC1* mRNA targeting operates independently from [+]-box-mediated mRNA docking. In the well-studied example of *ASH1* mRNA targeting to the bud tip in yeast, three steps can be distinguished: mRNA particle formation (which requires the She2 and She4 proteins), mRNA transport into the bud (which requires the She1, Myo4, and She3 proteins), and finally mRNA anchoring at the bud tip (which requires the She5, Bni1, Bud6, and Aip3 proteins) (*Beach and Bloom, 2001*). By analogy, Ire1 clusters may contain additional 'anchoring' factors for *HAC1* mRNA. The putative *trans*-acting factors postulated to bind the 3′ BE *cis*-acting element and guide *HAC1* mRNA to Ire1 clusters and those to mark the clusters and receive the incoming traffic remain unknown, as is the mechanism by which they would respond to ER-stress.

Ire1 clustering and splicing of its mRNA substrate are conserved in metazoan cells (*Yoshida et al., 2001*; *Calfon et al., 2002*; *Li et al., 2010*), but mRNA recruitment via a 3′ BE is not. Rather, *XBP1* mRNA (the metazoan ortholog of *HAC1* mRNA) is targeted to the ER membrane via a hydrophobic signal encoded in the unspliced mRNA (*Yanagitani et al., 2009*). This mechanism limits diffusion of the mRNA substrate to the two-dimensional plane of the membrane and thus may present an evolutionary bypass for docking via the [+]-box, functionally equivalent to the synthetic U1A bypass we describe here. Alternatively, membrane targeting in metazoans may represent the first targeting step that precedes a more specific docking event. Intriguingly, plants may employ both strategies: the unspliced substrate mRNA encodes a protein containing a hydrophobic membrane anchor (*Deng et al., 2011*), allowing pre-recruitment to the ER-membrane, while the conserved basic linker motif in plant Ire1 homologs may serve as a dedicated mRNA docking site for further selectivity in the recruitment process.

The step-wise reaction in which mRNA targeting and docking are staged and intertwined with Ire1 activation may help ascertain that Ire1 recognizes *HAC1* mRNA as its privileged substrate, even though at least 50 consensus sequence splice site motifs are present in the yeast transcriptome (*Gonzalez et al., 1999*; *Niwa et al., 2005*). Both in *Drosophila* and human cells, IRE1 processing of mRNA targets other than *XBP1* mRNA under prolonged ER-stress has been reported as part of an RNA-degradative pathway presumed to reduce the load of proteins entering the ER, called regulated Ire1-dependent decay (RIDD) (*Hollien and Weissman, 2006*; *Han et al., 2009*; *Hollien et al., 2009*). In *S. pombe*, which lacks *HAC1*, ER-stress is mitigated by Ire1 exclusively through RIDD and targets primarily mRNAs that encode proteins destined to enter the ER (*Kimmig et al., 2012*). In view of our data, RIDD may be invoked through modulating the stringency of mRNA delivery to and/or docking onto Ire1 clusters and may be precluded in *Saccharomyces cerevisiae* due to the stringency of the step-wise process. The organization of the engagement between Ire1 and its mRNA substrate(s) into multiple steps, that is targeting, docking, priming, and cleavage, thus emerges as integral to upholding selectivity and efficiency of Ire1-mediated mRNA processing and, hence, of UPR signaling.

## Materials and methods

### Yeast strains and plasmids

Standard cloning and yeast techniques were used for construction, transformation, and integration of plasmids and construction of yeast strains (*Sambrook et al., 1989*; *Longtine et al., 1998*; *Guthrie and Fink, 2002*). All mRNA visualization constructs as well as the *ire1Δ* strain, containing a genomic U1A–GFP copy or not, have been described (*Aragón et al., 2009*). Ire1 variants in all assays were expressed under the control of the autologous promoter at near-endogenous levels either from centromeric pRS315 (*Sikorski and Hieter, 1989*) derivatives or from a genomic copy integrated from pRS305 (*Sikorski and Hieter, 1989*) derivatives. Insertion of monomeric yeast-codon-adapted GFP or mCherry modules in the linker and IF[L], IF2[C], IF1[C], KD, and RD mutants of Ire1 have been described (*Papa et al., 2003*; *Lee et al., 2008*; *Aragón et al., 2009*; *Korennykh et al., 2009*; *Korennykh et al., 2011*). The IF3[C] mutant was created from the R899A mutant that was described before (*Korennykh et al., 2009*) to contain an additional substitution, K678A, which, based on the contacts in the crystal (*Korennykh et al., 2009*) would further eliminate interactions along that interface. The ΔR mutant was truncated

from P982, the ΔKR from L673, and the ΔKR/Δ[+]-box from K642. In the Δ[+]-box mutant residues K642–K658 and in the Δ55 mutant residues I579–E633 were deleted. All positively charged residues (K & R) within the K642–K658 stretch were replaced by glutamines in the Q-box mutant, or each K or R individually by threonine, while in the all [+] mutant the non-positively charged residues within the same stretch were replaced with arginines and lysines. The GFP, mCherry, or U1A modules were placed into the αF–αEF loop between H875 and S876 of Ire1 containing a S878T substitution that resulted from the cloning strategy. All Ire1 variants were constructed to contain a C-terminally encoded HA-tag. For the 'bypass' experiment (*Figure 5C*), *HAC1* mRNA variants were expressed under the control of the autologous promoter at near-endogenous levels from centromeric pRS316 (*Sikorski and Hieter, 1989*) derivatives. All yeast strains used for this study were based on the W303a derived CRY1 strain (*Aragón et al., 2009*; *Pincus et al., 2010*), including the newly constructed strains *ire1Δ*::KAN/*SpR*::HIS (used for all SpR splicing assays), *ire1Δ*::KAN/*ire1ΔKR-mCherry*::LEU, *ire1Δ*::KAN/*ire1ΔKR/Δ[+]box-mCherry*::LEU (used in *Figure 3B*), and *ire1Δ*::KAN/*hac1Δ*::HIS (used in *Figure 5C*). The SpR copy was integrated into the genome of the *ire1Δ* strain from a pRS304 (*Sikorski and Hieter, 1989*) plasmid derivative of pDEP005 (*Pincus et al., 2010*). The mCherry-tagged *IRE1* and *ire1* (ΔKR/Δ[+]-box) copies were integrated from pRS305 (*Sikorski and Hieter, 1989*) based constructs. All constructs used in this study are listed in *Supplementary file 1*.

## Growth conditions and stress induction

Cells were grown in standard or 2× concentrated synthetic media containing glucose as carbon source. Stress was induced either with DTT or tunicamycin, using concentrations at which differences between (samples from) wild-type and UPR deficient yeast are best appreciated, as we empirically established before: 0.2 µg ml$^{-1}$ tunicamycin for viability assays, 5 mM DTT for imaging of Ire1 clusters and mRNA recruitment (*Aragón et al., 2009*), and 2 mM DTT for the splicing reporter assay (*Pincus et al., 2010*).

## Splicing reporter assay

Two days after transformation of SpR harboring *ire1Δ* yeast (*ire1Δ*::KAN/*SpR*::TRP) with plasmids bearing Ire1 variants, fresh colonies were resuspended in 2× synthetic media each in 500 µl in 1-ml deep 96-well plates and incubated for 8 hr at 30°C. Fluorescence of samples was then analyzed either before or after the addition of 2 mM DTT and a further 2 hr incubation at 30°C by flow cytometry using a BD LSR-II, as described (*Pincus et al., 2010*).

## qRT-PCR assay

Total RNA was isolated by the hot phenol method and RT-PCR of spliced and total mRNA of *HAC1* and derivatives was performed, as described (*Elizalde et al., 2014*). The primers used to amplify spliced and total HAC1 mRNA were (forward) CTTGACAATTGGCGTAATCCAGAA (for spliced) and (forward) CCACGAAGACGCGTTGACTTGCAG (for total) and (reverse) GCTATATCGTCGCAGAGTGGGTCTG (for both spliced and total).

## Protein analysis

Protein extraction, electrophoresis, and transfer to nitrocellulose for immunoblot analysis of Ire1 variants with anti-HA antibody (12CA) were performed as before (*Pincus et al., 2010*). Endogenous Ire1 levels are at ~250 molecules/cell (*Ghaemmaghami et al., 2003*). In this study Ire1 variants were expressed at near-endogenous levels under the control of the autologous promoter from centromeric plasmids, and thus signals for HA-tagged Ire1 variants in immunoblots are weak.

## Microscopy and image analysis

All imaging and quantitation of images were performed, as described (*Aragón et al., 2009*). In brief, samples for microscopy were taken from yeast that was kept in early log-phase for at least 16 hr in 2× synthetic media before imaging. Microscopy of laser-excited mCherry or GFP was performed with a Yokogawa CSU-22 spinning disc confocal on a Nikon TE2000 microscope, controlled with µmanager and ImageJ. Images were captured with a 100×/1.4 NA Plan Apo objective on a Cascade II EMCCD and selected for analysis to contain significant signal above background but no saturated pixels. For display, images were processed in ImageJ and Adobe Photoshop such that the linear range of signals was comparable. Foci of Ire1 variants were determined and the co-localization index for U1A–GFP decorated SpR$^{U1A}$, SL-*PGK*-3' *hac1*$^{U1A}$, or SpR Δ3' BE$^{U1A}$ mRNA recruited to those foci was scored by using a customized MatLab script, as described (*Aragón et al., 2009*).

## Acknowledgements

We thank K. Thorn for his assistance with microscopy at the Nikon Imaging Center at UCSF and members of the Walter lab for discussion and comments. We dedicate this paper to the memory of Teresa Donovan. She will be sorely missed.

## Additional information

### Funding

| Funder | Grant reference number | Author |
|---|---|---|
| Nederlandse Organisatie voor Wetenschappelijk Onderzoek (Netherlands Organisation for Scientific Research) | Rubicon Postdoctoral Fellowship | Eelco van Anken |
| Giovanni Armenise-Harvard Foundation | Career Development Award | Eelco van Anken |
| Howard Hughes Medical Institute | | Peter Walter |
| National Science Foundation | Graduate Student Fellowship | David Pincus, Scott Coyle |
| Deutsche Forschungsgemeinschaft | Postdoctoral Fellowship | Christof Osman |
| Jane Coffin Childs Memorial Fund for Medical Research | | Alexei V Korennykh |
| Ministerio de Economía y Competitividad (Ministry of Economy and Competitiveness) | | Silvia Gómez Puerta |

The funders had no role in study design, data collection and interpretation, or the decision to submit the work for publication.

### Author contributions

EvA, Conception and design, Acquisition of data, Analysis and interpretation of data, Drafting or revising the article, Contributed unpublished essential data or reagents; DP, Acquisition of data, Analysis and interpretation of data, Contributed unpublished essential data or reagents; SC, Conception and design, Acquisition of data, Analysis and interpretation of data, Contributed unpublished essential data or reagents; TA, AVK, Analysis and interpretation of data, Contributed unpublished essential data or reagents; CO, Assisted with the viability assays; FL, Assisted with immunoblotting; SGP, Performed RT-PCR assays; PW, Conception and design, Analysis and interpretation of data, Drafting or revising the article

## Additional files

### Supplementary file

• Supplementary file 1. Overview of all constructs used in this study.

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
