## [Decision Letter]

Thank you for sending your work entitled “Specificity in endoplasmic reticulum stress signaling entails a step-wise engagement of *HAC1* mRNA to Ire1 clusters” for consideration at *eLife*. Your article has been favorably evaluated by Randy Schekman (Senior editor) and 3 reviewers, one of whom is a member of our Board of Reviewing Editors.

The Reviewing editor and the other reviewers discussed their comments before we reached this decision, and the Reviewing editor has assembled the following comments to help you prepare a revised submission.

All three comment on the significance of the work and mark it as a worthwhile advance. The review process also uncovered several deficiencies in your paper that should be remedied before publication. These are listed below.

The authors state: “Surprisingly, we found that residual targeting of HAC1 mRNA occurs even in the absence of recognizable features conserved among Ire1 proteins at the cytosolic side of the membrane (ire1ΔKR/Δ[+]-box), as long as the mRNA contained the 3' BE motif and was translationally repressed.” It is debatable if dependence on the 3'BE and on translational repression has been rigorously demonstrated. It would be important to show that the residual targeting is diminished when Hac1 mRNA lacks the 3'BE.

Figure 5 shows that the U1A-mediated targeting of Hac1 to Ire1 can partially compensate for the loss of the (+) box in Ire1 (comparing lanes 4 and 5). But even in this context the 3'BE is doing something to enhance splicing (lanes 5, 6). Given the focus of the paper on the interplay of RNA and protein motives in Hac1 mRNA targeting, it would be important to establish if U1A targeting can also compensate for the loss of the 3'BE: what does the splicing/tuni sensitivity look like for delta-3'BE when the U1A is not present in Ire1? Is splicing completely blocked (as with the delta+ box?)?

mRNA reporter assays and foci formation assays were done in the presence of 2 mM or 10 mM DTT while survival/growth assay were done in the presence of tunicamycin. Authors should provide the survival assays in the presence of DTT and foci formation and mRNA reporter assay in the presence of tunicamycin to determine if the phenomena is specific to DTT or is a general mechanism of ER stress resolution. Alternately, authors should provide a reasonable explanation why such experiments are not necessary.

The rationale for using various different ER inducers at different concentrations: tunicamycin (0.2 µg/ml) for Figure 1; 2 mM DTT for Figure 1; and 10 mM of DTT for Figure 1 for different times of incubation needs to be explained.

---

## [Author Response]

*The authors state: “Surprisingly, we found that residual targeting of HAC1 mRNA occurs even in the absence of recognizable features conserved among Ire1 proteins at the cytosolic side of the membrane (ire1ΔKR/Δ[+]-box), as long as the mRNA contained the 3' BE motif and was translationally repressed.” It is debatable if dependence on the 3'BE and on translational repression has been rigorously demonstrated. It would be important to show that the residual targeting is diminished when Hac1 mRNA lacks the 3'BE*.

The reviewers are correct that the present study does not rigorously demonstrate the dependence on the 3'BE and translational repression. However, we demonstrated this fact rigorously in our previous publication (Figures 2 and 4 in [1]). Consistent with these previous findings, in Figure 5 of the present study we observed no mRNA recruitment even for WT Ire1 in the absence of the 3'BE. Since recruitment is already down to background levels for WT Ire1, it is not plausible that *ire1* ΔKR/Δ[+]-box might recruit an mRNA lacking the 3'BE. We have further clarified this point in the text.

Figure 5
*shows that the U1A-mediated targeting of Hac1 to Ire1 can partially compensate for the loss of the (+) box in Ire1 (comparing lanes 4 and 5). But even in this context the 3'BE is doing something to enhance splicing (lanes 5, 6). Given the focus of the paper on the interplay of RNA and protein motives in Hac1 mRNA targeting, it would be important to establish if U1A targeting can also compensate for the loss of the 3'BE; what does the splicing/tuni sensitivity look like for delta-3'BE when the U1A is not present in Ire1? Is splicing completely blocked (as with the delta+ box?)?*

The reviewers are correct that this is a very important point. In fact, we performed the exact experiment the referees suggest in our previous publication (Figure 1 in [1]): a strain expressing WT Ire1 and *HAC1* mRNA lacking the 3’BE has a growth defect in ER stress conditions compared to WT. The reduction in splicing efficiency we now see upon removal of the 3’BE with the U1A ‘bypass’ is commensurate with the reduction seen for WT Ire1 upon removal of the 3’BE from *HAC1* mRNA. We now clarify and stress this point in the text.

*mRNA reporter assays and foci formation assays were done in the presence of 2 mM or 10 mM DTT while survival/growth assay were done in the presence of tunicamycin. Authors should provide the survival assays in the presence of DTT and foci formation and mRNA reporter assay in the presence of tunicamycin to determine if the phenomena is specific to DTT or is a general mechanism of ER stress resolution. Alternately, authors should provide a reasonable explanation why such experiments are not necessary*.

*The rationale for using various different ER inducers at different concentrations: tunicamycin (0.2 µg/ml) for*
Figure 1*; 2 mM DTT for*
Figure 1*; and 10 mM of DTT for*
Figure 1
*for different times of incubation needs to be explained.*

This point is well taken. Importantly, in [1] Figure S2, Figure 3, we monitored clustering in the presence of tunicamycin and DTT, and we showed that both stressors induce similar foci, though with different kinetics. There are three reasons we use 5 mM DTT for imaging and biochemical assays and 0.2 μg/ml tunicamycin for survival assays:

1) Since tunicamycin affects only newly synthesized proteins as it blocks *de novo* N-glycosylation, it takes longer to induce ER stress than DTT, which breaks disulfide bonds of already-synthesized ER clients. Thus, for assessing foci formation DTT is more suitable.

2) For the survival assay, strong ER stress will not reveal differences in UPR signaling efficiency. If we were to use 5 mM DTT in plate assays even the WT would be dead. The survival is measured after 3 days, whereas the clustering is measured after 1 hr. The 0.2 μg/ml tunicamycin concentration has been used widely in the field as it has been empirically established as a concentration at which ER stress is tolerable to WT cells but toxic to UPR mutants.

3) DTT in plate survival assays is not ideal as it oxidizes over time unless the medium is kept at a low pH. Tunicamycin in contrast is stable over longer periods of time. Using stabilized DTT would impose an acidic shock on top of the ER stress.

We now stress in the Methods that we established empirically that 5 mM DTT gives robust ER stress that is ideal for assessing clustering and mRNA recruitment, while 0.2 μg/ml tunicamycin is most suitable to score how mutants in UPR signaling efficiency affect growth.

Dr Peter Rodgers adds:

“We feel that the following title would be more understandable:

Specificity in endoplasmic reticulum stress signaling entails a step-wise engagement of HAC1 mRNA to clusters of the stress sensor Ire1”

We have changed the title according to Peter Rodgers’ excellent suggestion.